# Validating a model of architectural hazard visibility with low-vision observers

**Siyun Liu**[1]*, **Yichen Liu**[1], **Daniel J. Kersten**[1], **Robert A. Shakespeare**[2], **William B. Thompson**[3], **Gordon E. Legge**[1]

**1** Department of Psychology, University of Minnesota, Minneapolis, Minnesota, United States of America, **2** Department of Theatre, Drama, and Contemporary Dance, Indiana University Bloomington, Bloomington, Indiana, United States of America, **3** School of Computing, University of Utah, Salt Lake City, Utah, United States of America

* liux4433@umn.edu

**Data Availability Statement:** The data that were obtained in the experiment and supported the analysis in this paper were deposited in the Data Repository for the University of Minnesota (DRUM). The dataset contains the DeVAS

## Abstract

Pedestrians with low vision are at risk of injury when hazards, such as steps and posts, have low visibility. This study aims at validating the software implementation of a computational model that estimates hazard visibility. The model takes as input a photorealistic 3D rendering of an architectural space, and the acuity and contrast sensitivity of a low-vision observer, and outputs estimates of the visibility of hazards in the space. Our experiments explored whether the model could predict the likelihood of observers correctly identifying hazards. In Experiment 1, we tested fourteen normally sighted subjects with blur goggles that simulated moderate or severe acuity reduction. In Experiment 2, we tested ten low-vision subjects with moderate to severe acuity reduction. Subjects viewed computer-generated images of a walkway containing five possible targets ahead—big step-up, big step-down, small step-up, small step-down, or a flat continuation. Each subject saw these stimuli with variations of lighting and viewpoint in 250 trials and indicated which of the five targets was present. The model generated a score on each trial that estimated the visibility of the target. If the model is valid, the scores should be predictive of how accurately the subjects identified the targets. We used logistic regression to examine the correlation between the scores and the participants' responses. For twelve of the fourteen normally sighted subjects with artificial acuity reduction and all ten low-vision subjects, there was a significant relationship between the scores and the participant's probability of correct identification. These experiments provide evidence for the validity of a computational model that predicts the visibility of architectural hazards. It lays the foundation for future validation of this hazard evaluation tool, which may be useful for architects to assess the visibility of hazards in their designs, thereby enhancing the accessibility of spaces for people with low vision.

## Introduction

The accessibility of architecture determines how easily and safely its users can travel through its space and use its functional features. Visual accessibility determines whether vision can be

generated HVS of all subjects, and their responses being correct or incorrect in all the trials. The data is available through the following DOI: https://doi.org/10.13020/4h9x-xq26.

**Funding:** D.J.K, R.A.S, W.B.T, and G.E.L received funding from the National Institutes of Health, grant number EY017835. URL of NIH website: https://www.nih.gov/ The funders had no role in study design, data collection and analysis, decision to publish, or preparation of the manuscript.

**Competing interests:** The authors have declared that no competing interests exist.

used effectively and safely for mobility in an architectural space [1]. Designing visually accessible spaces for people with low vision is an objective with great significance. In the US, there were approximately 5.7 million people with uncorrectable low vision in 2017, and the number is expected to grow to 9.6 million by 2050 [2]. Worldwide, the number of people with moderate to severe visual impairment that fit the definition of low vision is estimated to be 217 million in 2015, and the number is predicted to reach 588 million by the year 2050 [3].

Increasing the visual accessibility of spaces for people with low vision is important for helping them maintain mobility and independence, hence improving their quality of life. To achieve this goal, it would be helpful to provide architects with tools to evaluate the visual accessibility of a space in the early stage of design, before it is brought to construction. The current report comes from an interdisciplinary project named Designing Visually Accessible Spaces (DeVAS). A major goal of the project is to develop a software tool for architects to assess hazard visibility in their designs.

The hazards in architecture are obstacles that could impede traveling through a plausible path. Obstacles can pose safety issues, such as tripping and falling, or bumping into things. Features like steps, stairs, benches and posts could be considered as hazards if not visible to pedestrians. The software tool developed by the DeVAS project estimates and visualizes hazard visibility for specified levels of reduced visual acuity (VA) and contrast sensitivity (CS). The visibility can be predicted by the available luminance contrast at the points in an image that correspond to actual depth and orientation changes of surfaces in the scene [4].

This software differs in two fundamental ways from existing practice in architectural design. First, the software employs principles of *luminance-based design* rather than *illumination-based design*. Typically, architecture uses illumination-based design in which lighting standards are stipulated in terms of overall light flux falling on surfaces measured in lux. Illumination-based design ensures that there is sufficient lighting. But visibility of a hazard depends on variations in luminance across the scene: more specifically, on the viewing location of the observer, the angular size of the hazard, its contrast with the background, and also the vision status of the observer. We refer to design that takes these factors into account as Luminance-based design. Our approach is novel in employing luminance-based design. Second, our approach explicitly takes reduced vision (low vision) into account in evaluating the visibility of architectural hazards. It does so by including information about the acuity and contrast sensitivity of observers. No existing software used by architects or lighting designers includes explicit reference to the vision status of people with low vision. The goal of this paper is to describe empirical studies aimed at validating this novel software approach to architectural design. Our experiments compared the accuracy of human observers with reduced acuity and contrast sensitivity in recognizing step hazards ahead with the predictions of the software.

The software is described by Thompson et al. [5] and available at https://github.com/visual-accessibility/DeVAS-filter. In brief, the software takes the following inputs: a 3D computer-aided design (CAD) model and light sources of an architectural space, the vision parameters of a sample human observer (VA and CS) and the observer's viewpoint in the space. The workflow of the software is illustrated in Fig 1.

The first step is to render a 3D simulation of the space from the desired viewpoint (Fig 1A). Because the visibility of a hazard from the user's viewpoint depends on its visual angular size and luminance contrast, we need to use a photometrically accurate method to render a perspective image from the 3D model. We used the Radiance rendering software for this purpose [6]. The rendered high dynamic range image contains accurate luminance values of the designed space under the specified light sources.

Essential information includes the observer's vision status and viewpoint. In principle, inclusive design aims at providing accommodation for the widest possible range of vision

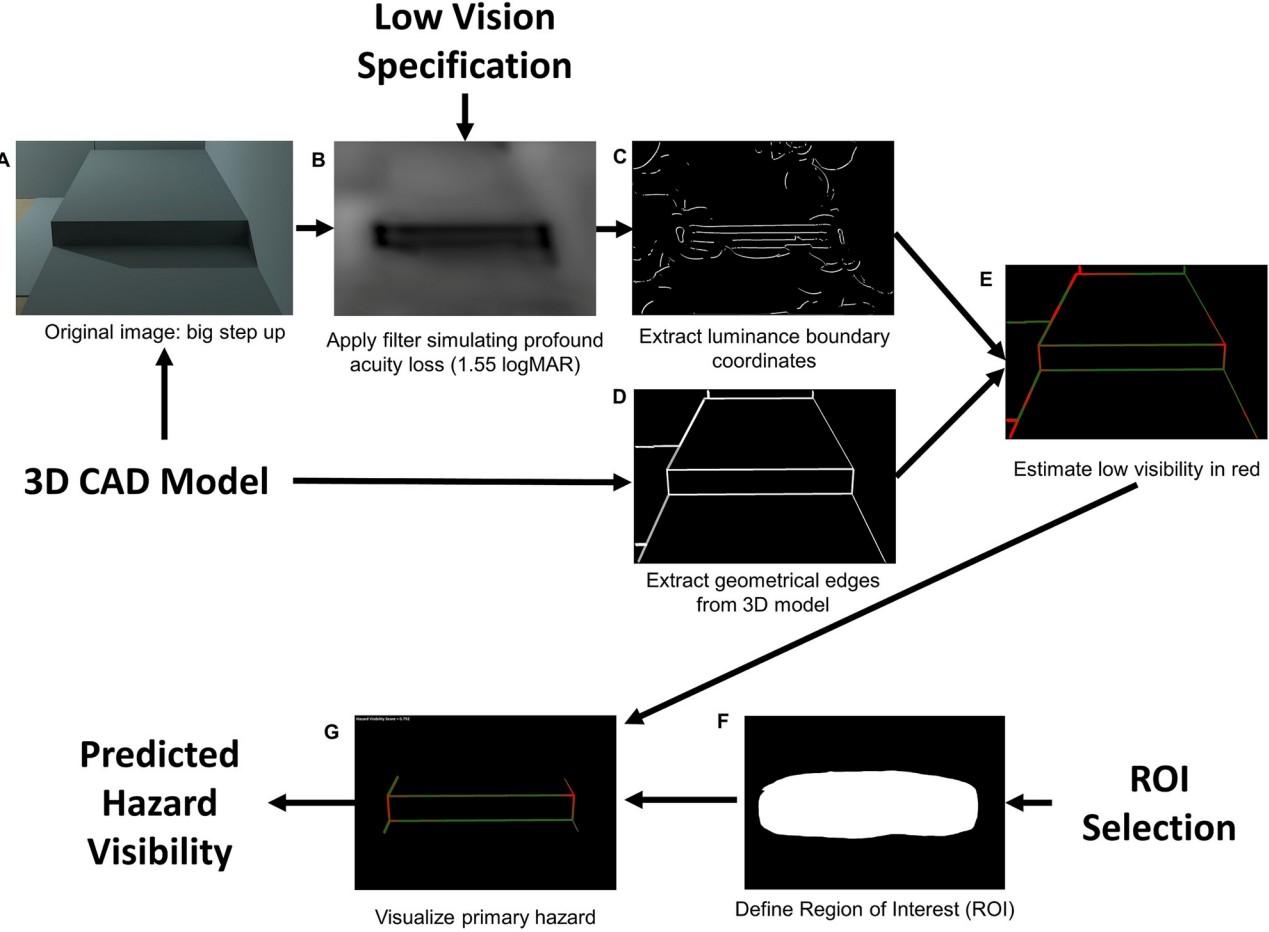

**Fig 1. Workflow of the DeVAS software.** A: the original image of a step rendered by Radiance software. B: image filtered to simulate Severe low vision (VA 1.55 logMAR, CS 0.6 Pelli-Robson). C: luminance boundaries extracted from the filtered image. D: Pixels representing Geometric edges inferred from 3D data map of the space. E: Estimation of hazard visibility, based on a match between the luminance contours in C and the geometrical edges in D. Color coding represents the closeness of the match, ranging from red (poor match) to green (good match). F: A manually defined Region of Interest (ROI), G: The conjunction of E and F which specifies the hazard region of primary focus, which is used to generate the final Hazard Visibility Score (HVS).

conditions. In practice, a designer may wish to specify values of VA and CS and viewing distance to test a certain hazard's visibility. For example, is a step visible at 10 feet for a person with VA of 20/400 or better? The next step in the computational flow is to represent the reduced VA and CS of a potential low-vision observer. Chung & Legge [7] have shown that for many people with low vision, the contrast sensitivity function (CSF) has the same shape as the CSF for normal vision but shifted leftward on the log spatial-frequency axis and downward on the log contrast-sensitivity axis.

With VA and CS specified, a CSF is generated, which gives contrast threshold as a function of spatial frequency. The software uses the CSF to threshold the image with a non-linear filter so that only the visual information above the perceptual threshold of the sample low-vision pedestrian will be kept [8, 9]. An example of a filtered image to simulate acuity VA of 1.55 log-MAR and CS of 0.6 Pelli-Robson is presented in Fig 1B.

The next step in the software flow is to extract luminance boundaries from the filtered image, using Canny edge detection [10]. These boundaries mark pixel locations in the image with high intensity change, and which represent edges likely visible to the low-vision pedestrian (panel C).

The software uses the 3D CAD representation of the architecture to locate the geometrical edges of each object in the image in terms of pixel coordinates (Fig 1D). Visibility of the geometrical edges is estimated by the adjacency (match) between a geometrical edge and a luminance feature. For each pixel on the geometrical edges, we calculate how close it is (in number of pixels) to its nearest neighbor on the luminance boundaries. We use the separation as an indicator of visibility. The pixel separation is transformed to a 0–1 score (See formula in S1 Appendix), where a higher score means smaller separation and higher visibility, and lower score means larger separation and a lower visibility. In Fig 1E, a red-green color code is used where red means low visibility (more dangerous) and green means high visibility (safer).

The software user can define a Region of Interest (ROI) so that the software can assess the visibility of target features within the ROI, instead of everything in the image (Fig 1F). The average score across all pixels on the geometrical edges in the ROI is the feature's Hazard Visibility Score (HVS) (Fig 1G).

The HVS measure of visibility is based on matching low-level luminance cues in an image to geometrical features. However, there are two major reasons why this score might not correspond to perceptual judgments made by low-vision observers. First, the method represents low vision only by including measures of clinical VA and CS in the filter. Variability in low-vision performance is likely to depend on additional factors including characteristics of visual-field loss and diagnosis-specific factors. Second, human observers are likely to use other information in addition to contrast features of stimuli in making judgments about hazard identification. Top-down information, such as prior expectations and contextual cues are likely to play a role.

We conducted two experiments to determine if our Hazard Visibility Score (HVS) has predictive power on judgments made by human observers. In Experiment 1, normally sighted subjects with artificially reduced acuity were tested with Radiance-generated images on a calibrated computer display. They were asked to distinguish between large and small stepping hazards under conditions of varying lighting and viewpoint. For each test image and subject, we used the DeVAS software to generate an HVS. We assessed the validity of the HVS by testing whether the performance accuracy of our subjects was significantly related to the HVS. To link the continuous predictor (HVS) with binary responses (correct or incorrect identification), we used logistic regression for the analysis.

Previous studies with normally sighted subjects wearing artificial acuity reduction have shown similar patterns of dependence on environmental variables as studies with low-vision subjects with comparable acuity [1, 11, 12]. For this reason, we conducted our first experiment with simulated visual impairment to determine whether our task requirements and stimuli were sufficiently challenging for a range of reduced acuities and contrast sensitivities, and to ensure a sufficient spread in HVS and performance scores to avoid floor and ceiling effects. When the first experiment demonstrated the viability of the protocol, we conducted Experiment 2 with ten low-vision subjects using the same stimulus set.

In summary, the HVS model predicts the visibility of architectural features based on their contrast and spatial-frequency content, taking the contrast sensitivity and acuity of the observer into account. The experiments described in this paper test the idea that visibility, computed in this way, plays a role in the recognition of architectural hazards. Our experimental results support this view.

## Methods

### Subjects

The experiment followed a protocol approved by the University of Minnesota IRB. Each subject signed an IRB-approved consent form.

In Experiment 1, there were 22 normally sighted adults (10 males and 12 females) recruited from the University of Minnesota at Twin Cities campus. The mean age of the subjects was 21.3 years, with a standard deviation of 1.29 years. One subject dropped out in the middle of the experiment, so their data were discarded.

The remaining 21 subjects were assigned to three conditions: Normal (no blur), Moderate blur, and Severe blur. Each condition contained 7 subjects. We used the Lighthouse Distance Visual Acuity chart to measure the subject's acuity (VA) and the Pelli-Robson chart (1-meter viewing distance) to measure contrast sensitivity (CS).

The no-blur group did the experiment with their corrected to normal vision. Their correct identification rates were used to ensure that the stimuli were reliably visible to normally sighted viewers. We used diffusive films to create two levels of artificial acuity reduction, termed Moderate and Severe. For the Moderate group, we used 2 layers of Rosco Roscolux 132 sheet gel to reduce the mean VA to 1.2 logMAR (SD 0.085) and mean CS to 0.68 (SD 0.1). For the Severe group, we used 1 layer of Rosco Roscolux 140 sheet gel and the mean VA was 1.62 logMAR (SD = 0.028), mean CS 0.6 (SD 0.019). For the Severe group, the Pelli-Robson chart was inappropriate to measure their contrast sensitivity because the chart's angular print size at the 1-meter viewing distance is too small for the reduced acuity. We used a formula derived from past data to infer their contrast sensitivity from acuity [13]. The formula is:

$$CS = 1.72 - 0.69 * VA, R2 = 0.51.$$

In Experiment 2, the subjects were 10 adults (4 females, 6 males) with diverse forms of low vision. Subject age, gender, VA, CS, and diagnosis are shown in Table 1. We tested one additional pilot low-vision subject to evaluate the experimental protocol. Their data was not analyzed. For Subjects 8 and 9, whose VA was worse than 1.5 logMAR, we used the formula mentioned above to estimate CS.

Our primary consideration in recruiting low-vision subjects was to secure a range of VA and CS in the Moderate to Severe range, without regard to diagnostic categories. Prior to recruiting, we verified that VA and CS parameters in this range would yield a widely distributed spread of HVS for our set of test images. Subjects with milder low vision would likely

**Table 1. Low-vision subject information.**

| Subj No. | Age | Gender | Acuity | Contrast Sensitivity | Diagnosis |
|---|---|---|---|---|---|
| LV1 | 66 | F | 0.8 | 1.65 | macular hole |
| LV2 | 40 | F | 1.28 | 0.6 | Aniridia |
| LV3 | 31 | M | 1.14 | 0.3 | retinitis pigmentosa |
| LV4 | 54 | M | 1.16 | 1.05 | Aniridia |
| LV5 | 21 | F | 1.5 | 0.3 | aniridia, glaucoma, nystagmus |
| LV6 | 58 | M | 1.36 | 0.8 | congenital cataract |
| LV7 | 38 | F | 1.44 | 0.2 | retinitis pigmentosa |
| LV8 | 60 | F | 1.54 | 0.65 | familial vitreo-retinopathy, cataract |
| LV9 | 56 | M | 1.66 | 0.57 | optic nerve atrophy |
| LV10 | 45 | F | 1.02 | 1.55 | glaucoma, congenital cataract, degenerative myopia |

Geometry, Lighting, and Viewpoint Variation of Stimuli

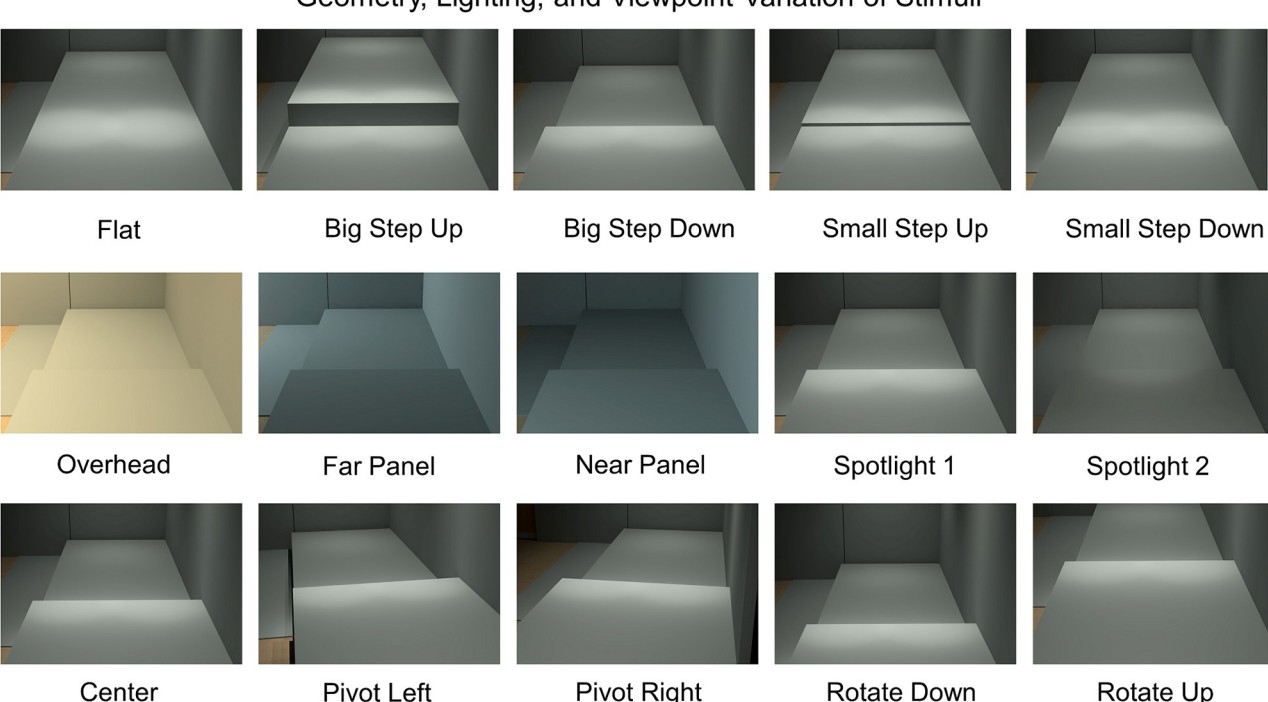

**Fig 2. Geometry, lighting, and viewpoint variation of stimuli.** The top row used the lighting setting "spotlight 1" and viewpoint setting "center" to demonstrate the five target types: flat, big step-up big step-down, small step-up, and small step-down. The middle row used big step-down and center viewpoint to show the five lighting variations: overhead, far panel, near panel, spotlight 1, and spotlight 2. The bottom row used big step-down and spotlight 1 to show the five viewpoints: center, pivot left, pivot right, rotate down, and rotate up.

have had corresponding HVS scores near ceiling, weakening our validation test. Similarly, subjects with more severe low vision would likely have shown floor effects.

## Stimuli

The stimuli in both experiments were computer-generated images showing a 30 feet long walkway with one of the five possible targets: big step-up, small step-up, big step-down, small step-down, and flat. Big steps are seven-inch high and small steps are one-inch high. The targets are the same width as the walkway, which is four feet. We used steps as our targets because they are a common and potentially dangerous type of hazard for people with low vision. We matched the reflectance of ground materials so that the luminance on each part of the image would be in accordance with the luminance of corresponding places in the original classroom. Lighting and viewpoint each had five variations. Fig 2 shows examples of the five targets, five lighting arrangements, and five viewpoints. These stimulus images are combined with two artificial blur levels (Moderate 1.2 logMAR and Severe 1.6 logMAR) to generate HVS values for experimental trials. Fig 3 shows in each blur condition, how many trials fall in each bin of width 0.1 spanning the HVS range from zero to one.

Radiance software was used to render the test images from accurate 3D representations of a space. The images show an architectural space based on a campus classroom.

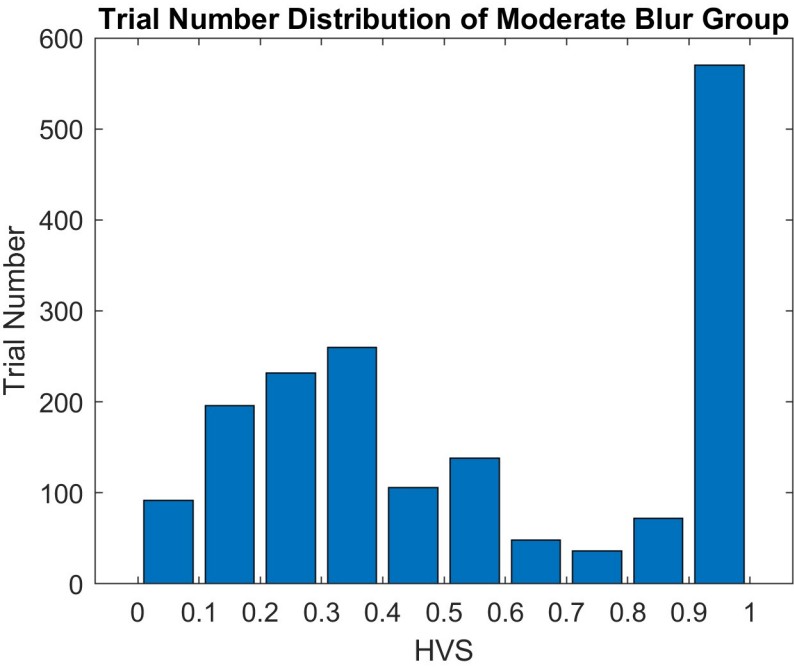

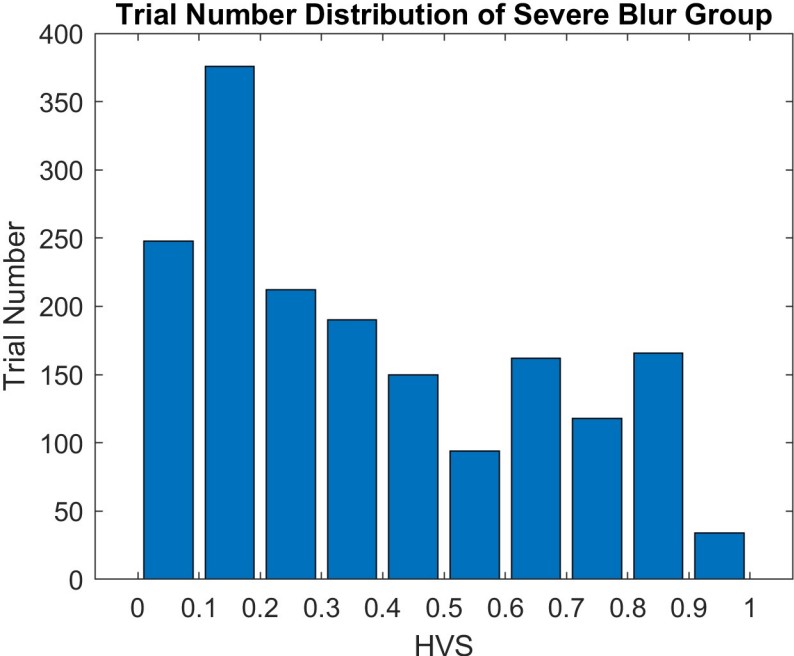

**Fig 3. Distribution of trials in ten HVS bins, each covering 0.1 width in a zero to one range.** The upper panel shows the trial distribution for Moderate blur (1.2 logMAR) and the lower panel shows the distribution for Severe blur (1.6 logMAR).

## Apparatus

We used an NEC E243WMi-BK 16:9, 24" widescreen monitor. The stimuli were presented with Matlab PsychToolbox [14]. The actual room and corresponding Radiance rendering have a larger dynamic range than the monitor. However, previous work [15] has shown that calibrating the screen to match the ratio of the luminance across real boundaries provides screen displays matching the luminance contrast in the real classroom. The rendered images were created with a virtual viewpoint ten feet from the target steps. The 7-inch step height viewed from 10 ft subtended approximately 3.3º and the one-inch step height subtended approximately 0.45º. The step width subtended 23º. To match the visual angles, the participant sat at 32 inches away from the screen. We measured the eye-to-screen distance once before each experiment started.

## Procedure

For each subject, we measured acuity and contrast sensitivity with their normal correction, and for the normally sighted subjects, with the blur goggles they were assigned to wear. The resulting estimates of reduced VA and CS were used as inputs to the software to generate HVS values for each image and each subject.

In pilot testing, we discovered that subjects with low vision had more difficulty than sighted subjects to understand the context of our stimuli. To ensure the subjects understood the rendered room layout, we made a small tactile model out of Legos for the low-vision subjects to touch to help them understand the spatial layout of the simulated testing space. We also showed and explained sample images to them and provided them with practice trials. We made sure they were familiar with the five targets, and the variations in lighting and viewpoint.

Each trial consisted of the presentation of a stimulus image followed by the subject's response. For subjects with artificially reduced acuity, the presentation time was one second. A pilot experiment with a low-vision subject, whose data was not used in analysis, indicated that low-vision subjects would sometimes require more time to scan the image and make a decision. Therefore, with the low vision subjects, the presentation time was two seconds. Subjects made two responses on each trial. First, they indicated which of the five targets was present (five-alternative forced choice). They then gave a confidence rating on a one-to-five scale, with one meaning pure guessing, and five meaning highly confident. Results from the confidence ratings will not be reported in this article. The experimenter registered answers through the keyboard. The subject started the next trial at their own pace by clicking the mouse.

There were 250 (5 targets* 5 lighting * 5 viewing angle * 2 repetitions) trials in total. Each subject viewed and responded to all 250 images. The stimuli were presented in a randomized order. Responses were not timed. The whole procedure took from one to two hours. Each participant completed all the trials within one session.

## Data analysis

The hazard visibility score (HVS) was calculated for each image and each subject, taking the subject's VA and CS as parameters in the filter component of the DeVAS software workflow. The HVS score ranges from zero (no visibility) to one (maximum visibility) and was computed with the formula given in S1 Appendix. We used the HVS values of stimulus images as the independent variable and the subject's correct or incorrect response for each trial as the dependent variable. To assess the association between recognition accuracy and HVS, we fitted a logistic regression with aggregated data from all subjects in a group, as well as for each subject individually. We used logistic regression because it assesses the relationship between a continuous predictor (HVS score) and a binary outcome (correct or incorrect response). This was a

convenient method for our design in which there were many trials with different HVS values across subjects and conditions, each scored with a binary outcome (correct or incorrect identification). The model can be described by the following equation:

$$Ln\left(\frac{P}{1-P}\right) = A * X + B \tag{1}$$

Where X refers to the HVS score and P represents probability of a correct response.

For purposes of plotting in Figs 4, 6–8, we rearrange the equation to plot P vs. X:

$$P = \frac{\exp^{A*X+B}}{\exp^{A*X+B} + 1} \tag{2}$$

The fitted regression model contains two parameters, a slope (A) and an intercept (B). The slope, A, is an indicator of the relationship between the predictor and the odds (P/(1-P)) of the event being predicted. If the slope is significantly larger than zero (p value less than .05), there is a statistically significant positive relationship between proportion correct and the HVS value.

In logistic regression models, the predictive power of HVS can be assessed by both the magnitude of slope and the goodness-of-fit metric. The greater is the slope, the stronger is the correlation between identification accuracy and the HVS score. The goodness-of-fit is measured by comparing the deviance of an intercept-only null model and the fitted model with HVS added as predictor. Deviance measures the difference between each dependent variable observation and the predicted value of the model. The difference between the null deviance and the residual deviance represents how much predictive power the independent variable adds to the model. To make cross-subject comparisons, we use the reduced deviance ratio to quantify goodness-of-fit, which is the ratio of the difference between null and residual deviance divided by the null deviance. The higher the ratio, the better the fitted model predicts the data, hence stronger the HVS' predictive power.

An ANOVA test is also run against each fitted logistic regression model and an intercept-only null model to see if adding HVS as a predictor significantly improves the ability to estimate subjects' identification accuracy.

We used the glmer function in lme4 (version 1.1–23) package of R (version 3.6.0) for accumulated data to account for random effects and used glm function for individual subject data [16, 17]. The ANOVA test was conducted by calling anova.glm function, and the test type was Chi-square.

## Results

### Experiment 1—Performance of normally sighted subjects with artificial acuity reduction

Three groups of normally sighted subjects were tested, one group without blur and the other two with goggles that artificially reduced acuity. The average performance accuracy of the normally sighted group with no blur on all trials was 98.23%, close to 100%, confirming that the step hazards were recognizable for people with normal vision.

We accumulated data across subjects in the two blur groups separately. There were 1750 (7 subjects * 250 trials) datapoints for each blur condition. Both groups had slopes significantly higher than zero, showing that there is a statistically significant positive correlation between HVS and percent correct. For the Moderate blur group, the slope was 3.02, meaning that for a unit increase in HVS, the logarithm of the odds of making correct response over incorrect response increases 3.02 times. For the Severe blur group, the slope was 1.54 (p < .001). Fig 4

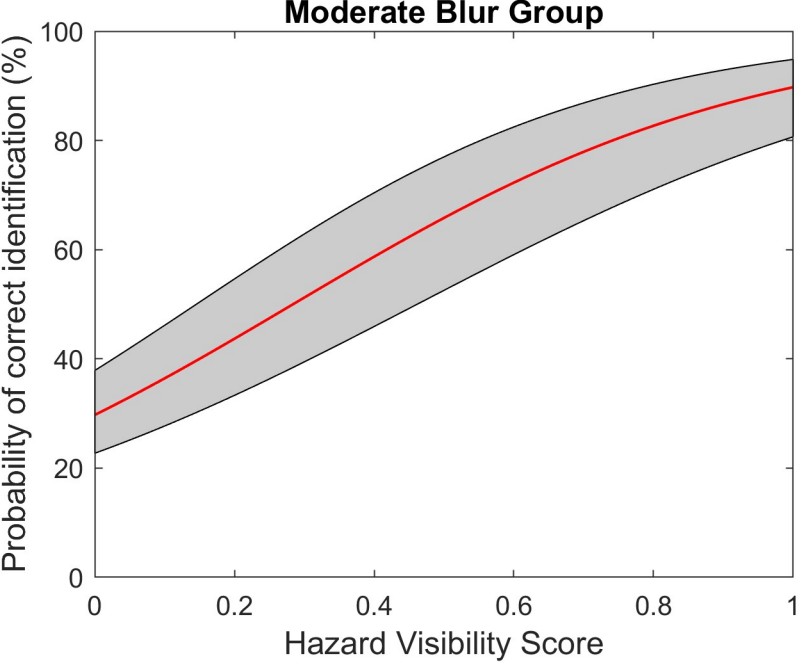

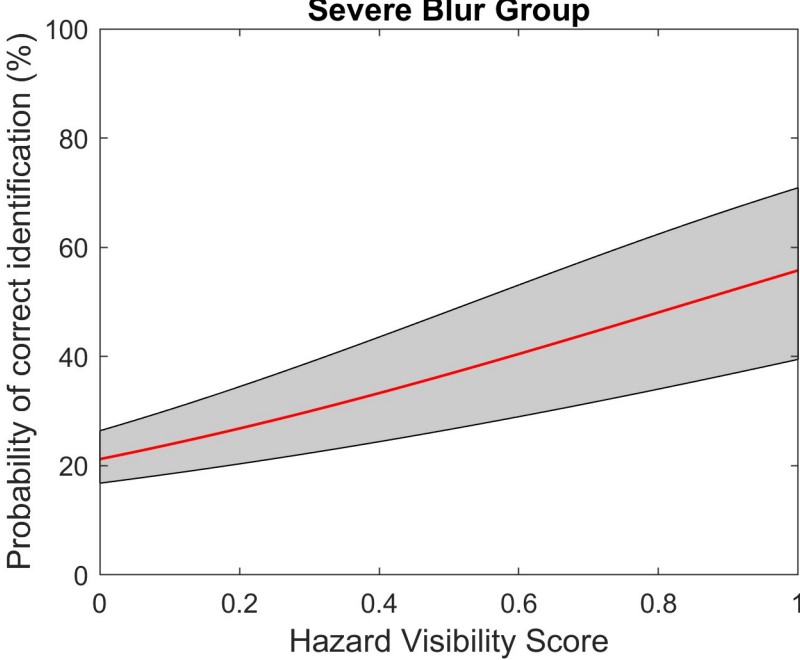

**Fig 4. Logistic regression model of aggregated data from subjects viewing with artificial blur.** Top: Moderate Blur (seven subjects), mean acuity 1.2 logMAR. Bottom: Severe Blur (seven subjects), mean acuity 1.6 logMAR. The red line shows the logistic regression function, transformed as shown in Eq 2. the gray area represents 95% confidence intervals.

shows the estimated successful identification probability curve plotted from the fitted logistic regression models. A Chow test showed that the two models are significantly different ($F$ $(2,3496) = 125.97$, $p < .001$).

In order to investigate the difference between the slope values of the Moderate and Severe blur groups, we looked at the number of correct and incorrect trials in each 0.1 interval of HVS from zero to one. The result is presented as the histogram in Fig 5. Percent correct plateaued near 50% for the Severe group, remaining at this relatively low level for HVS values

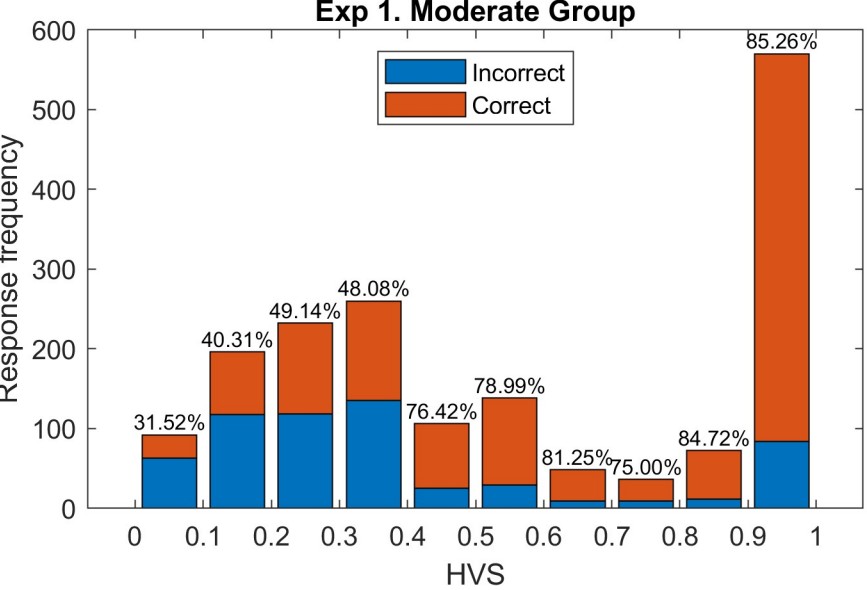

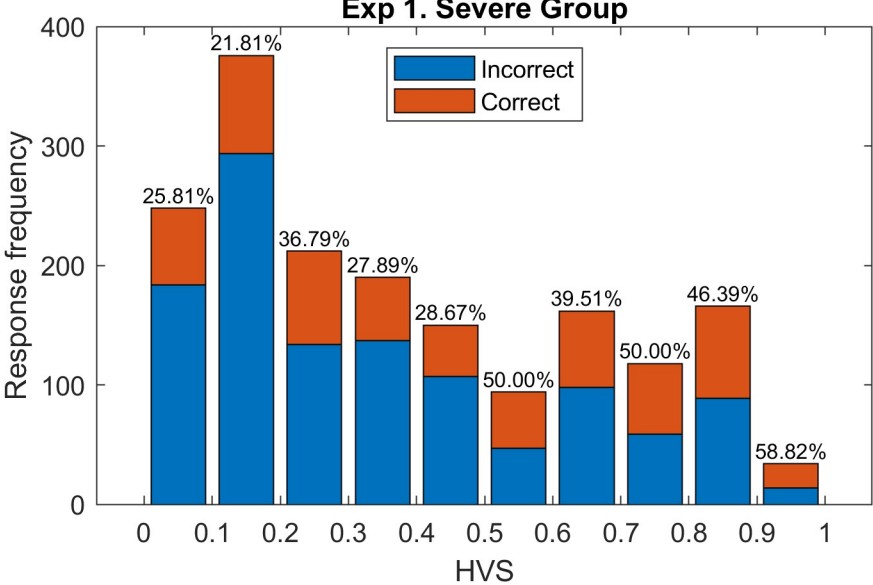

**Fig 5. Histograms presenting correct and incorrect trials in each 0.1-wide bin of the HVS accumulated across seven subjects in each blur group.** The upper panel shows the distribution for moderate blur group trials, and the lower panel shows the distribution of severe blur group trials.

above 0.5. Apparently, factors not captured by the HVS score held down performance in these trials.

We also fitted logistic-regression models for each subject in the two blur groups individually, shown in Fig 6. 12 out of 14 subjects had slopes significantly higher than zero, meaning

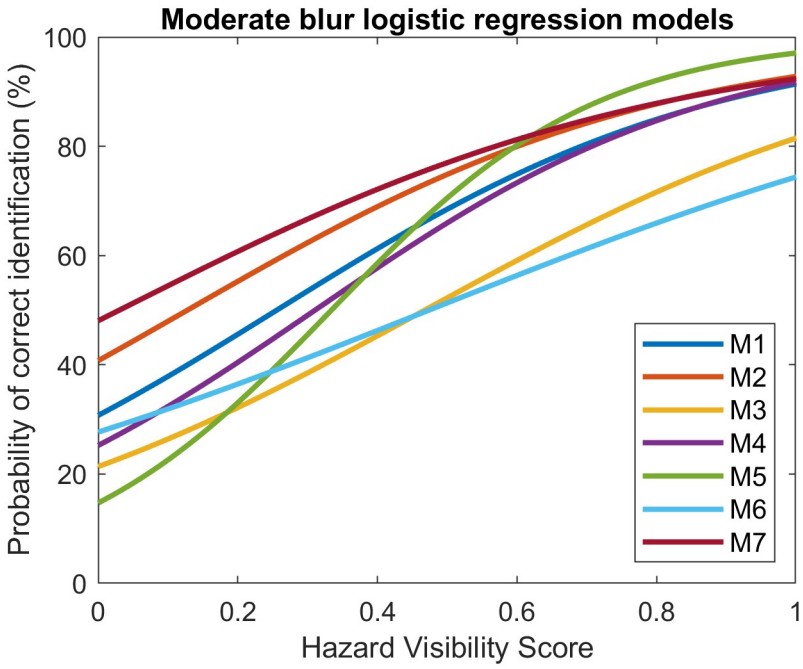

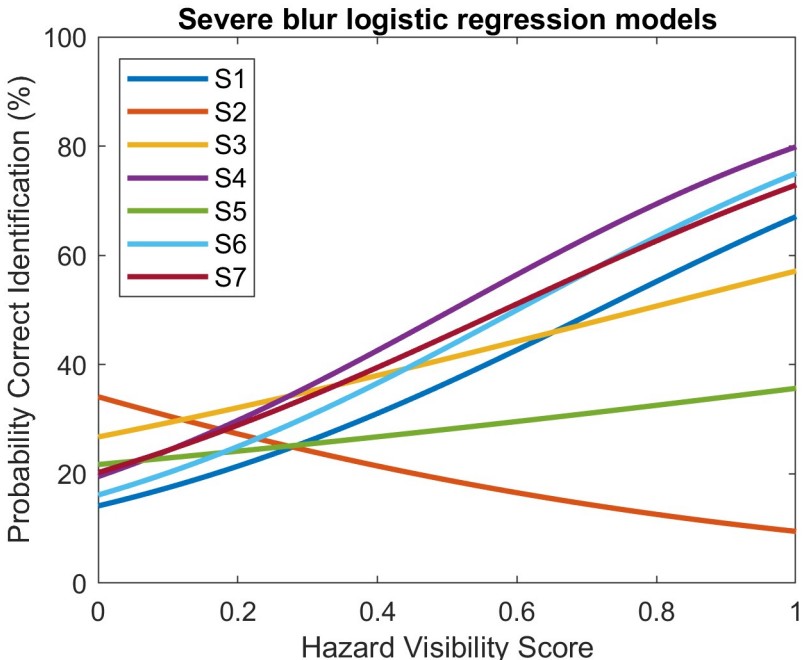

**Fig 6. Logistic regression models of 14 subjects with artificial acuity reduction by blur.** Top: Moderate Blur Group. Bottom: Severe Blur Group.

**Table 2. Individual regression models for subjects in moderate and severe blur groups.**

| Subject ID | Blur Group | Slope | Slope Confidence Interval | Null Deviance | Residual Deviance | Reduced deviance ratio |
|---|---|---|---|---|---|---|
| M1 | Moderate | 3.17*** | [2.13, 4.21] | 313.43 | 267.8162 | 15% |
| M2 | Moderate | 2.93*** | [1.87, 3.98] | 298.3458 | 259.9361 | 13% |
| M3 | Moderate | 2.78*** | [1.91, 3.65] | 341.3715 | 295.8234 | 13% |
| M4 | Moderate | 3.49*** | [2.44, 4.54] | 319.1733 | 263.9916 | 17% |
| M5 | Moderate | 5.25*** | [3.85, 6.66] | 310.3458 | 214.9572 | 31% |
| M6 | Moderate | 2.02*** | [1.19, 2.85] | 345.2765 | 320.4507 | 7% |
| M7 | Moderate | 2.57*** | [1.45, 3.68] | 284.4157 | 259.8404 | 9% |
| S1 | Severe | 2.51*** | [1.44, 3.58] | 310.34 | 287.49 | 7% |
| S2 | Severe | -1.6** | [-2.74, -0.45] | 265.9621 | 257.8065 | 3% |
| S3 | Severe | 1.29** | [0.38, 2.20] | 332.0321 | 324.0198 | 2% |
| S4 | Severe | 2.79*** | [1.78, 3.810] | 340.146 | 307.3867 | 10% |
| S5 | Severe | 0.69*** | [-0.27, 1.66] | 290.6297 | 288.6757 | 1% |
| S6 | Severe | 2.74*** | [1.65, 3.84] | 325.541 | 299.2504 | 8% |
| S7 | Severe | 2.36*** | [1.33, 3.38] | 333.9225 | 312.1075 | 7% |

**: P-Value < .005.

***: P-value < .001.

their odds of making correct identification increased with HVS. The slopes are presented in Table 2.

There are two outliers in the Severe Blur group (S2 and S5), one having non-significant slope, the other a negative slope. From their confusion matrices, we found these two subjects mistook most big-step-up trials for either small-step-up or big-step-down. Since the big-step-up trials had high HVS, mistakes on these trials lowered their performance on the high HVS end, hence affecting the slopes.

Table 2 includes the slopes as well as Null and Residual Deviance and the reduced deviance ratio of all 14 logistic regression models in both blur groups. Although the reduced deviance ratio was generally lower than 30%, the ANOVA tests showed that for most subjects, HVS significantly improved prediction compared with the null model (indicated by asterisks in Table 2).

## Experiment 2—Performance of low-vision subjects

The logistic model for the aggregated data of ten low-vision subjects is shown in Fig 7. The model is based on 2500 trials, 250 trials for each of the ten subjects. It had a slope of 3.45 (p < .001).

Individually fitted logistic models also had slopes significantly larger than zero for all ten subjects, as shown in Fig 8.

Table 3 lists the Null Deviance and Residual Deviance for each individual model, along with the reduced deviance ratio. All individual models fitted for low-vision subjects were significantly better than the null model.

For the low-vision subjects in Experiment 2, we also found that the predictive power of HVS values was weaker for subjects with poorer acuity and contrast sensitivity. There was a negative correlation between the regression slope values and the logMAR acuities (slope equals -6.25, $R^2 = 0.35$) and a positive correlation with Pelli-Robson contrast sensitivities (slope equals 4.12, $R^2 = 0.56$), both indicating that the HVS score was a better predictor for more moderate

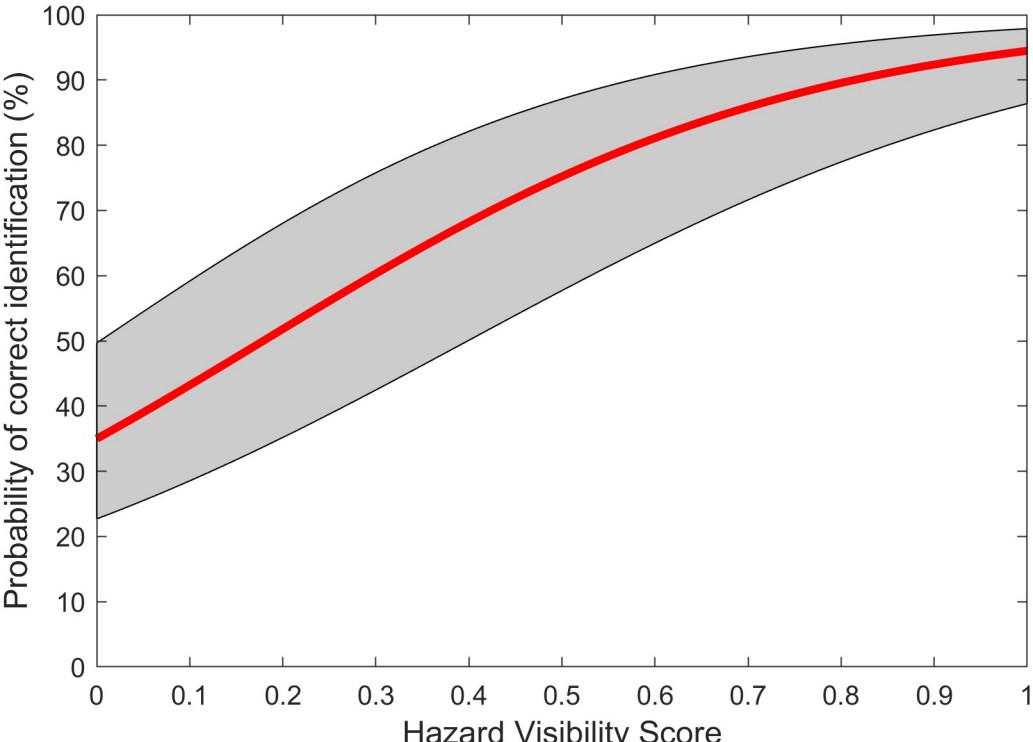

**Fig 7. Logistic regression model of aggregated data from low-vision subjects.** The red line shows the regression curve and the gray area outlines the upper and lower bounds of the 95% confidence interval of the slope and intercept.

(less severe) vision loss. Fig 9 shows the scatterplots of logistic regression slope values plotted against acuity and contrast sensitivity. Regression lines are also presented.

## Discussion

### Does HVS predict human performance with reduced acuity?

As discussed in the Introduction, there were two issues challenging the validity of the Hazard Visibility Score (HVS) as a predictor of human performance in identifying architectural hazards First, whether using only VA and CS could effectively capture an observer's ability in perceiving architectural hazards. Second, whether estimating visibility of 3D geometrical boundaries of hazards could effectively predict an observer's ability to identify the hazard. We found that the HVS estimation of architectural hazard visibility is associated with human performance in identifying hazards in viewers with low vision as well as normally sighted observers with artificially reduced acuity. There was a consistent positive correlation between the HVS and correct identification rate, indicated by the regression slopes. The regression model's slope varied from individual to individual, yet all low-vision subjects and 12 out of 14 normally sighted subjects with artificially reduced acuity had slopes significantly larger than zero.

These findings provide a first step in validating the approach of assessing architectural feature visibility using the computational model implemented in the DeVAS software and described by Thompson and colleagues [5].

However, there is still substantial residual deviance in the data, meaning that the HVS did not successfully predict subjects' identification in many trials. This indicates that HVS alone does not fully account for human performance on identifying architectural features.

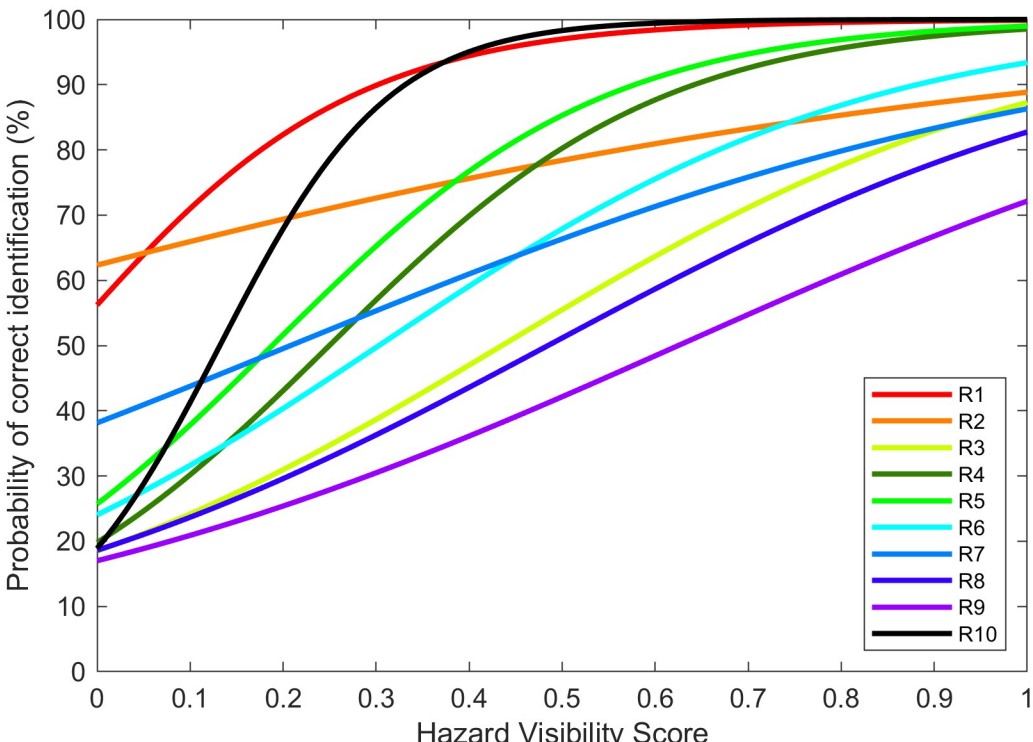

**Fig 8. Logistic regression models of 10 low-vision subjects.**

We can identify some factors that might affect an observer's perception of architectural features not accounted for by the HVS. Patterns of field loss associated with different diagnostic categories such as central loss from macular degeneration or peripheral loss from glaucoma might affect performance by influencing the portion of the scene viewed. A related issue is the difference in strategies adopted by our subjects. For example, whether a person makes eye movements to explore visual space outside their restricted visual field impacts their performance in detecting obstacles in the environment [18, 19].

**Table 3. Individual regression models of low-vision subjects.**

| Subject ID | Slope | Slope Confidence Interval | Null Deviance | Residual Deviance | Reduced deviance ratio |
|---|---|---|---|---|---|
| LV1 | 6.45** | [1.91, 10.99] | 70.81353 | 52.82475 | 25% |
| LV2 | 1.57** | [0.56, 2.57] | 265.9621 | 255.7521 | 4% |
| LV3 | 3.23*** | [2.29, 4.16] | 337.3001 | 276.5907 | 18% |
| LV4 | 5.60*** | [4.00, 7.19] | 265.9621 | 178.1817 | 33% |
| LV5 | 5.56*** | [3.87, 7.24] | 338.7891 | 267.4427 | 21% |
| LV6 | 3.79*** | [2.68, 4.90] | 326.7091 | 266.4454 | 18% |
| LV7 | 2.27*** | [1.15, 3.39] | 346.5096 | 327.496 | 5% |
| LV8 | 3.04*** | [2.05, 4.02] | 344.266 | 302.3751 | 12% |
| LV9 | 2.53*** | [1.44, 3.62] | 324.338 | 302.1003 | 7% |
| LV10 | 11.03*** | [5.25, 16.81] | 124.2173 | 72.18844 | 42% |

**: P-Value < .005.

***: P-value < .001.

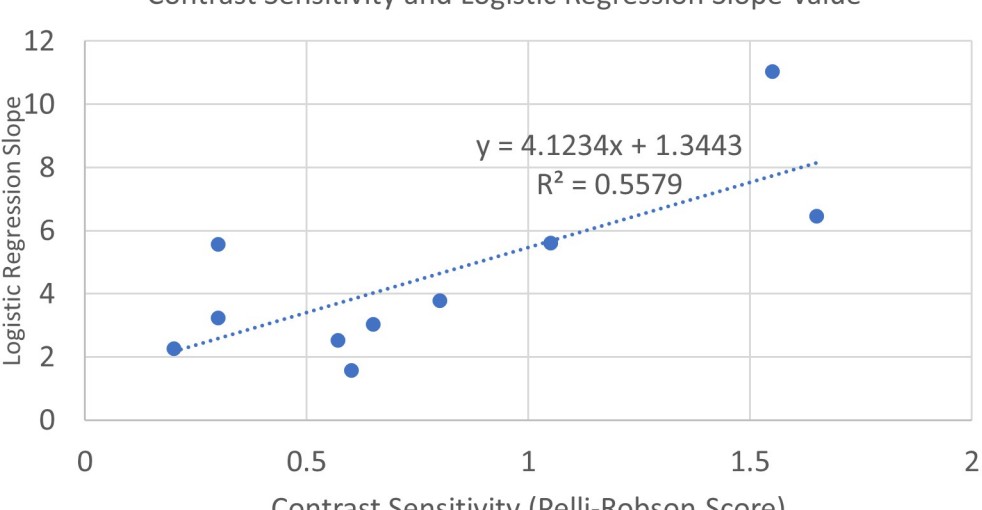

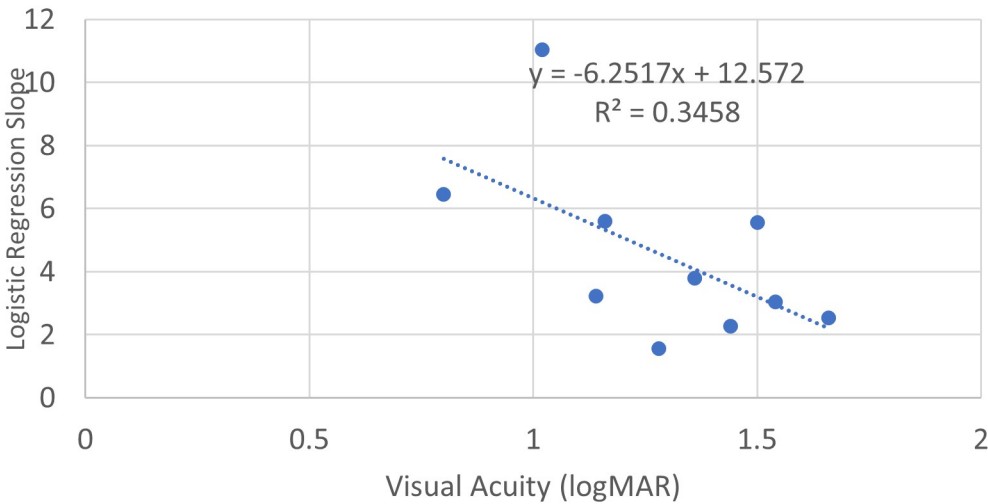

**Fig 9. A scatterplot of logistic regression slope values of individual subjects and their visual acuities (upper panel) and contrast sensitivities (lower panel).** Linear regression trend lines are also plotted.

## Visual impairment severity and HVS predictive power

Our data from both experiments indicate that the HVS score had greater predictive power for moderate vision loss compared with more severe vision loss. These results may indicate that the HVS is more useful for a range of moderate vision impairment lying between normal vision and the most severe forms of vision loss approaching total blindness.

In Experiment 1 with normally sighted subjects, the group with Severe blur made many errors, even for stimuli with high HVS scores, predicted to be highly visible. Most of the stimuli with high HVS values were images of the large step up. Examination of the confusion

matrix for trials with above-0.8 HVS values(shown in S2 Appendix) indicated that these images were often confused with the small step up or the large step down. It is possible that the relevant geometric boundaries were visible but the subjects were not able to interpret the 3D meaning of these features.

### HVS predictive power contingent on ROI

It is important to be aware that the HVS is dependent on how the user defines the Region of Interest (ROI). The HVS is an average of visibility estimates along all 3D geometrical contours within the ROI, so its value will depend on the geometry within the selected ROI.

An example drawn from the current experiment is shown in Fig 10. In this lighting condition, the contrast is high only on the left and right edges of the step, while the horizontal edge between has very low contrast. In this case, defining the ROI as only the left and right corners, or only the horizontal edge between the corners, or the whole step yields different values of HVS.

We tested whether changing the ROI definition influences the predictive power of the HVS. We fitted logistic regression models with identification responses as dependent variable and the HVS generated with the central edge ROI as the independent variable. The central edge ROI is demonstrated in Fig 10, the third panel from left. S3 Appendix contains the model statistics and visualization. With the ROI changed from the complete region to the central edge, the slope magnitude and reduced deviance ratio increased for 5 out of the 10 subjects.

The stronger association between the HVS score for the central-edge-only ROI may imply that these subjects relied on this cue rather than the corner features. If so, their attention to this cue may have been due to a deliberate strategy or might have been related to visual-field restrictions.

Currently, the DeVAS software provides two types of information about the visibility of hazards. One type is an imagery visualization as demonstrated in Fig 1G. In the ROI, the more hazardous, or less visible parts of the geometry is colored in red, while the less hazardous, or more visible parts are colored in green. The second type is the numeric HVS representing the overall visibility of the geometrical boundaries within the ROI. An architect might look at the color-coded visualization to see the visibility of the local features within a broadly defined ROI and use the numerical HVS value as a summary statistic.

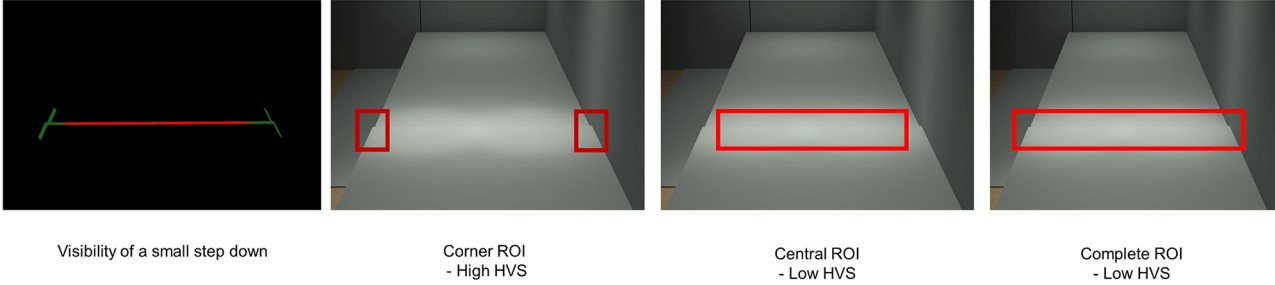

**Fig 10. ROI's influence on visibility estimation.** A small downward step is shown in original resolution with three different definitions of ROI. From left to right, the first panel visualizes the visibility of the step with VA equivalent to 1.15 logMAR and CS 0.85 Pelli-Robson. Within this step, the left and right corners (green part) have high visibility, whereas the central horizontal edge (red part) has low visibility. The ROI can be defined as the side corners (second panel), or the central horizontal edge (third panel), or the whole step (fourth panel). The HVS derived from the corners ROI is 0.889, central ROI 0.051, while the ROI based on the entire step had HVS of 0.106. The definition of ROI will often lead to a significant change in HVS.

## Limitations

We comment on two limitations of our study. First, the current project only looked at correlating HVS and the identification of a small set of well-defined hazards in an architectural space. However, pedestrians with low vision in real life often have to process a much more complicated space. The pedestrians may not know how many hazards they need to attend to, what types they are, and their possible locations. Existence of multiple hazards, distribution of attention, the location of the hazard in the visual field, and saliency of the hazards, may all influence their identification. Second, the low-vision subjects in this study were in a limited range of acuities (0.8 to 1.6 logMAR) and diagnostic categories. Future work will be necessary to determine the validity of the HVS across a wider spectrum of low vision.

## Conclusion

We have provided initial evidence for the validity of a computational model that estimates visibility of hazards in architectural spaces. Further work will be required to examine the general applicability of this computational model. We showed that the performance of human observers with artificially reduced acuity and a group of observers with low vision in identifying step hazards was related to an algorithmically generated numeric estimate of visibility called the Hazard Visibility Score (HVS). The HVS was based on a model taking into account a viewpoint-dependent photometrically accurate 3D rendering of a hazard in the visual field and an observer's visual acuity and contrast sensitivity. The method may be applied in architectural design to assess visibility of hazards, thereby enhancing the accessibility of spaces for people with low vision. While the ultimate validation of the DeVAS software will require it to be applied to a more diverse sample of architectural designs and a wider range of low-vision users, the current study was intended as a first step in validating the HVS metric as a measure of visibility for people with low vision.

## Supporting information

**S1 Appendix. Formula for HVS derivation.**
(DOCX)

**S2 Appendix. Experiment 1 severe blur group high-HVS trials confusion matrix.**
(DOCX)

**S3 Appendix. Individual regression models of low-vision subjects fitted with alternative (central only) ROI.**
(DOCX)

**S1 File. Data analysis R code markdown.**
(PDF)

## Acknowledgments

Thanks to Rachel Gage for her contribution in the data collection process. Thanks also to Professor Nathan Helwig and Miss Jiaqi Liu for their statistical advice.

## Author Contributions

**Conceptualization:** Daniel J. Kersten, William B. Thompson, Gordon E. Legge.

**Data curation:** Siyun Liu.

**Formal analysis:** Siyun Liu, Yichen Liu.

**Funding acquisition:** William B. Thompson, Gordon E. Legge.

**Investigation:** Siyun Liu, Yichen Liu.

**Methodology:** Daniel J. Kersten, Robert A. Shakespeare, William B. Thompson, Gordon E. Legge.

**Resources:** Robert A. Shakespeare, William B. Thompson.

**Software:** Robert A. Shakespeare, William B. Thompson.

**Supervision:** Daniel J. Kersten, William B. Thompson, Gordon E. Legge.

**Validation:** Siyun Liu, Yichen Liu, Daniel J. Kersten.

**Visualization:** Siyun Liu, Daniel J. Kersten, Robert A. Shakespeare, William B. Thompson.

**Writing – original draft:** Siyun Liu.

**Writing – review & editing:** Daniel J. Kersten, Robert A. Shakespeare, William B. Thompson, Gordon E. Legge.

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
