## [Decision Letter · Decision Letter 0]

25 May 2021

PONE-D-21-08870

Validating a model of architectural hazard visibility with low-vision observers

PLOS ONE

Dear Dr. Liu,

Thank you for submitting your manuscript to PLOS ONE. After careful consideration, we feel that it has merit but does not fully meet PLOS ONE’s publication criteria as it currently stands. Therefore, we invite you to submit a revised version of the manuscript that addresses the points raised during the review process.

Both reviewers find your paper interesting an valid. Reviewer 2 in particular provides several constructive comments which will further improve the presentation of your work. I expect it should be simple to address these comments, and I look forward to the revised version of your manuscript. 

We look forward to receiving your revised manuscript.

Kind regards,

Guido Maiello

Academic Editor

PLOS ONE

Journal Requirements:

2. We noted in your submission details that a portion of your manuscript may have been presented or published elsewhere.

"In a previous paper from our group (Thompson et al.), now in press at LEUKOS The Journal of the Illuminating Engineering Society, we described in detail our computational model of architectural hazards. I’m attaching a copy of our LEUKOS paper for you and the reviewers. It is cited in our current manuscript (reference 5). In the Thompson et al paper, we reported a highly abbreviated summary of the results of Experiment 2 from the current paper. In the current paper, we provide an abbreviated description of the computational model detailed in the Thompson paper, and  we provide a detailed description of our empirical validation of the model. we compare simulated low-vision (Experiment 1) with actual low-vision (Experiment 2), together with a full analysis of these data. There are no figures in common in the two papers. We believe that the two papers have substantially original an different content with only modest overlap."

Reviewers' comments:

Reviewer's Responses to Questions

**Comments to the Author**

1. Is the manuscript technically sound, and do the data support the conclusions?

Reviewer #1: Yes

Reviewer #2: Partly

2. Has the statistical analysis been performed appropriately and rigorously? 

Reviewer #1: Yes

Reviewer #2: Yes

3. Have the authors made all data underlying the findings in their manuscript fully available?

Reviewer #1: Yes

Reviewer #2: Yes

4. Is the manuscript presented in an intelligible fashion and written in standard English?

Reviewer #1: Yes

Reviewer #2: Yes

5. Review Comments to the Author

Reviewer #1: This is a well-written, interesting and novel paper which validates a computer model of visibility for architectural features (steps). They model takes visual acuity and contrast sensitivity into account but, as the authors acknowlege, visual field loss is not considered.

In future work I suggest that visual field loss (central or peripheral) is accounted for, as well as the effect of binocular vision and parallax, which is likely to affect the visibility of objects such as steps and kerbs.

Reviewer #2: Review: Validating a model of architectural hazard visibility with low-vision observers

The paper presents two experiments aimed to test a model that simulates the visibility of hazards in an architectural environment for people with low vision. The goal of this model is to drive the design of spaces to fit the visual abilities of people with low vision. The experiments show that the model estimates of visibility were able to predict performance of both people with simulated low vision and people with low vision. This is an interesting paper that has the potential to influence the design of architechtural spaces.

Main Points:

• It’s unclear how the model and the experiments translate to actual real-world visibility of steps. The manuscripts describes a few factors that can affect visibility in real world scenarios such as contextual cues and prior expectations. How does the HVS model relate to real world scenarios? I assume the model will be used to design real world spaces that are accessible for low vision people. This is briefly mentioned in the “limitations” section but deserves a more in depth description in intro and discussion.

• The experiment considers 5 architectural options (e.g., step up, step down, etc) and participants were given a chance to explore a lego model of these options. Yet, in a realistic scenario there are many more options than these five, and a low vision walker would not have the privilege to select only among 5 options. How and why were these spaces/steps chosen?

• Experimental procedure. There were only 2 repetitions per condition? Why? That seems low for an experiment. I could not find the timing of stimulus presentation and response.

• Analysis. Can you provide analysis for RT?

• Analysis/Figures:

o The number of trials in each HVS level is different (fig 3), but the graphs do not reflect it. Can you incorporate the number of trials as the size of dots in the graphs (Fig 4,6,7, etc)? Did different points get different weights according to the number of trials? Because with such a variability, the reliability of each point is likely to be different (especially with only 2 trials per conditions).

o In Figures 5, it is hard to estimate the percent correct for each HVS level. These points could be added to figure 4 which will also help see the fit of the line.

• How does lighting conditions affect HVS? Is there a better light source for each step type? Is it the same light source across step types?

* Missing confusion matrices for two groups. In the presented matrix, any ideas why participants did not correctly identify the big step down? How come "big down", "small up", and "small down" were so poorly identified?

Minor points:

• Figures were not numbered and were not of high quality. I assumed it had to do the submission process.

• X- Axes in figure 3 and 5 is not clear (values are 1-10, when they seem to represent 0-1 in steps of 0.1).

• Missing axes in fig 9.

6. PLOS authors have the option to publish the peer review history of their article (what does this mean?). If published, this will include your full peer review and any attached files.

Reviewer #1: **Yes: **Michael Crossland

Reviewer #2: **Yes: **Sarit Szpiro

---

## [Author Response · Author response to Decision Letter 0]

20 Sep 2021

Dear Editors,

We appreciate the reviewer’s generous comment on the manuscript! Here are our responses to each of the points in their comment. We also listed what we’ve changed in the manuscript in response to their concerns. 

Reviewer #2

Main Points:

1. It’s unclear how the model and the experiments translate to actual real-world visibility of steps. The manuscript describes a few factors that can affect visibility in real world scenarios such as contextual cues and prior expectations. How does the HVS model relate to real world scenarios? I assume the model will be used to design real world spaces that are accessible for low vision people. This is briefly mentioned in the “limitations” section but deserves a more in depth description in intro and discussion.

Answer: 

The model aims to assess the visibility of geometrical features of potential hazards in a space, such as the edges of a step, bench or post. The model assumes that if the geometrical feature is below the low-vision pedestrian’s detection threshold, based on its contrast and spatial-frequency content, the hazard may not be seen. We acknowledge in the manuscript that factors beyond threshold visibility can affect the ability of a low-vision pedestrian to see a hazard. For example, if the pedestrian has a restricted field and fails to include the hazard within the field of view, it may be missed, even if the hazard is above threshold when in the field of view. Also, contextual factors, such as the presence of a visible railing near a step, might alert a pedestrian about the presence of a step that is below threshold. We acknowledge that our model does not take factors other than threshold visibility into account, and our validation test is limited in its scope.

To address your comment, we made revisions in the Introduction section (p.4, line 30-34).

In summary, the HVS model predicts the visibility of architectural features based on their contrast and spatial-frequency content, taking the contrast sensitivity and acuity of the observer into account. The experiments described in this paper test the idea that visibility, computed in this way, plays a role in the recognition of architectural hazards. Our experimental results support this view.

2. The experiment considers 5 architectural options (e.g., step up, step down, etc) and participants were given a chance to explore a Lego model of these options. Yet, in a realistic scenario there are many more options than these five, and a low vision walker would not have the privilege to select only among 5 options. How and why were these spaces/steps chosen?

Answer:

Steps and other ground-plane irregularities are common hazards for people with low vision. We hope our software will be helpful to architects in designing steps and ramps with good visibility. From a practical point of view, we needed to conduct our validation study with a small set of potential targets. We felt that steps, with some variation in viewpoint, lighting, and size, would provide a reasonable approximation to real-world stepping hazards. In pilot testing of subjects with low vision, we discovered that they found it more difficult than sighted subjects to understand the context of our test images. For this reason, and to ensure clarity in the task, we showed them a tactile model of the general testing scenario. As mentioned in the Discussion, we acknowledge that there will be many cases in which people with low vision are unfamiliar with the potential hazards in a space, so this is a limitation of our study. 

To address this comment, we added a sentence in the Methods – Stimuli section (page 6, line 10-13):

We used steps as our targets because they are a common and potentially dangerous type of hazard for people with low vision. 

And a sentence in the Methods – Procedure section (page 7, line 10-14): 

In pilot testing, we discovered that subjects with low vision had more difficulty than sighted subjects to understand the context of our stimuli. To ensure that the subjects understood the rendered room layout, we made a small tactile model out of Legos for the low-vision subjects to touch to help them understand the spatial layout of the simulated testing space.

3. Experimental procedure. There were only 2 repetitions per condition? Why? That seems low for an experiment. I could not find the timing of stimulus presentation and response.

Answer: We designed the testing protocol so that the experiment could be completed within one session lasting approximately two hours. The tests included 250 trials for each participant, which required about two hours for most of our low vision subjects. 

Presentation time was one second for normally-sighted subjects (artificially reduced acuity), and two seconds for low-vision subjects. The information is presented in section Methods – Procedure, page 7, line 18 and 21: 

Each trial consisted of the presentation of a stimulus image followed by the subject’s response. For subjects with artificially reduced acuity, the presentation time was one second. A pilot experiment with a low-vision subject, whose data was not used in analysis, indicated that low-vision subjects would sometimes require more time to scan the image and make a decision. Therefore, with the low vision subjects, the presentation time was two seconds.

We also added in the manuscript that we did not time the participants’ response (page 7, line 29): 

Responses were not timed. The whole procedure took from one to two hours.

• Analysis. Can you provide analysis for RT? 

Answer:

We did not intend to measure and analyze RT (reaction time) in this study. Since we are testing the validity of our hazard-visibility estimation software, performance accuracy was the key dependent variable. We encouraged subjects to provide their judgment of the target in the image without any time pressure, which we felt was more representative of how they would make decisions in the real world. 

• Analysis/Figures:

4. The number of trials in each HVS level is different (fig 3), but the graphs do not reflect it. Can you incorporate the number of trials as the size of dots in the graphs (Fig 4,6,7, etc)? Did different points get different weights according to the number of trials? Because with such a variability, the reliability of each point is likely to be different (especially with only 2 trials per conditions).

Answer:

Figure 4, 6, 7 are the visualization of logistic regression curves we fitted from the experiment data. The HVS values are generated from the stimuli image and the observer’s visual acuity and contrast sensitivity. We presented 125 images to all the participants, while each participant had different acuities and contrast sensitivities. Therefore, only the two repeated trials tested with the same subject had the exact identical HVS values. In all the logistic regression models, each datapoint used to fit the model corresponds to one trial in the experiment, and all datapoints were treated equally in the statistical model. The logistic models are not directly related to the bars in Figure 3 and Figure 5. By the nature of logistic regression model, the skewness of the distribution of independent variable should not affect the validity of the fitted model. We didn’t find any extreme values or influential points in any of the logistic regression models, either. Therefore, we feel that adding the information in Figure 3 and Figure 5 to Figure 4, 6, and 7 might confuse and mislead the reader. 

5. In Figures 5, it is hard to estimate the percent correct for each HVS level. These points could be added to figure 4 which will also help see the fit of the line.

Answer: 

We added percent correct information in Figure 5. As explained in the answer to point #4, the logistic regression model visualized in Figure 4 was not fitted based on the accuracy of trials in each HVS levels, which was shown in Figure 5. We worry that integrating these two pieces of information would mislead the readers, hence prefer keeping the two figures separate. 

6. How does lighting conditions affect HVS? Is there a better light source for each step type? Is it the same light source across step types?

Answer:

Lighting conditions and the geometrical layout of the feature both affect HVS. Different light sources work differently on every step type, some make the step highly visible, and some make it very hard to identify. The effect of lighting conditions working together with step types can be seen in Figure 2. In general, the panel lightings (as in condition “far panel” and “near panel”) results in the best visibility for all step types, while the ranking of mean HVS by lighting conditions can be different from one step type to another. The purpose of this paper is to validate the visibility estimation of a software, but not to systematically investigate the effect of lighting on visibility. Therefore, in the manuscript, we did not include the interaction analysis between lighting and step type in our stimuli set. 

7. Missing confusion matrices for two groups. In the presented matrix, any ideas why participants did not correctly identify the big step down? How come "big down", "small up", and "small down" were so poorly identified?

Answer:

The confusion matrix in the manuscript is an explanation of why the correlation between HVS and subject identification accuracy is comparatively weaker in the severe blur group than in moderate blur group. Therefore, it only contained trials with above-0.8 HVS in the severe blur group of Experiment 1. We did not include the complete confusion matrix in the paper because we were focusing on the errors made on trials with high HVS values.

Some trials with images containing big step down, small step up, and small step down are misidentified, despite having above-0.8 HVS. The reason why this happened might be that the contour of the step was partially visible but could not indicate the correct step type and rule out other possibilities. Therefore, although the relevant geometric boundaries were visible, the subjects were not able to interpret the 3D meaning of these features. 

Minor points:

8. Figures were not numbered and were not of high quality. I assumed it had to do the submission process.

The figures were not numbered following the guidelines provided by Plos One. The website specified “Do not include author names, article title, or figure number/title/caption within figure files. That information will go into your figure caption in the manuscript.” 

We have changed an image export method, hoping it would improve the image quality shown in the submitted version. 

9. X- Axes in figure 3 and 5 is not clear (values are 1-10, when they seem to represent 0-1 in steps of 0.1).

We changed the X-axes labels of figure 3 and 5 to 0-1, with ten 0.1 intervals. 

10. Missing axes in fig 9.

We added X-axis labels in Figure 9.

---

## [Decision Letter · Decision Letter 1]

8 Nov 2021

Validating a model of architectural hazard visibility with low-vision observers

PONE-D-21-08870R1

Dear Dr. Liu,

We’re pleased to inform you that your manuscript has been judged scientifically suitable for publication and will be formally accepted for publication once it meets all outstanding technical requirements.

Kind regards,

Guido Maiello

Academic Editor

PLOS ONE

Additional Editor Comments (optional):

Reviewers' comments:

Reviewer's Responses to Questions

**Comments to the Author**

1. If the authors have adequately addressed your comments raised in a previous round of review and you feel that this manuscript is now acceptable for publication, you may indicate that here to bypass the “Comments to the Author” section, enter your conflict of interest statement in the “Confidential to Editor” section, and submit your "Accept" recommendation.

Reviewer #2: All comments have been addressed

2. Is the manuscript technically sound, and do the data support the conclusions?

Reviewer #2: Yes

3. Has the statistical analysis been performed appropriately and rigorously? 

Reviewer #2: Yes

4. Have the authors made all data underlying the findings in their manuscript fully available?

Reviewer #2: Yes

5. Is the manuscript presented in an intelligible fashion and written in standard English?

Reviewer #2: Yes

6. Review Comments to the Author

Reviewer #2: I thank the authors for addressing my comments, especially regarding Figs 4-6. The manuscript is ready for publication.

7. PLOS authors have the option to publish the peer review history of their article (what does this mean?). If published, this will include your full peer review and any attached files.

Reviewer #2: **Yes: **Dr. Sarit Szpiro, Special Education Department, University of Haifa, Israel

---

## [Editor Report · Acceptance letter]

12 Nov 2021

PONE-D-21-08870R1 

Validating a Model of Architectural Hazard Visibility with Low-Vision Observers 

Dear Dr. Liu:

I'm pleased to inform you that your manuscript has been deemed suitable for publication in PLOS ONE. Congratulations! Your manuscript is now with our production department. 

Kind regards, 

on behalf of

Dr. Guido Maiello 

Academic Editor

PLOS ONE